# Evaluation of Agricultural Machinery Operational Benefits Based on Semi-Supervised Learning

Yashuo Li, Bo Zhao *, Weipeng Zhang, Liguo Wei and Liming Zhou

State Key Laboratory of Soil Plant Machine System Technology, China Academy of Agricultural Mechanization Science Group Co., Ltd., Beijing 100083, China
* Correspondence: caamsjds309@gmail.com

**Abstract:** Judging the efficiency of agricultural machinery operations is the basis for evaluating the utilization rate of agricultural machinery, the driving abilities of operators, and the effectiveness of agricultural machinery management. A range of evaluative factors—including operational efficiency, oil consumption, operation quality, repetitive operation rate, and the proportion of effective operation time—must be considered for a comprehensive evaluation of the quality of a given operation, an analysis of the causes of impact, the improvement of agricultural machinery management and an increase in operational efficiency. In this study, the main factors affecting the evaluation of agricultural machinery operations are extracted, and information about the daily operations of particular items of agricultural machinery is taken as a data source. As regards modeling, a subset of data can be scored manually, and the remaining data is predicted after the training of the relevant model. With a large quantity of data, manual scoring is not only time-consuming and labor-intensive, but also produces sample errors due to subjective factors. However, a small number of samples cannot support an accurate evaluation model, and so in this study a semi-supervised learning method was used to increase the number of training samples and improve the accuracy of the least-squares support vector machine (LSSVM) training model. The experiment used 33,000 deep subsoiling operation data, 500 of which were used as training samples and 500 as test samples. The accuracy rate of the model obtained using 500 training samples was 94.43%, and the accuracy rate achieved with this method with an increased number of training samples was 96.83%. An optimal combination of agricultural machinery and tools is recommended owing to their operational benefits in terms of reduced costs and improved operating capacity.

**Keywords:** semi-supervised learning; operational benefits; agricultural machinery operations

## 1. Introduction

The operational efficiency of a piece of agricultural machinery refers to the amount of work completed by the item in question per unit of operating time. It is an important indicator of the utilization effect of agricultural machinery. Given the aim of ensuring operational quality, improving the operational efficiency of agricultural machinery can speed up the progress of work and reduce operating costs.

In the use of agricultural machinery, unreasonable selection of agricultural machinery, unscientific multi-machine marshalling, poor operational speed control, and low time utilization will affect operational efficiency [1–3]. The efficiency of agricultural machinery operations is not completely determined by the efficiency of agricultural machinery. The blind pursuit of high-power agricultural machinery and a lack of suitable agricultural machinery will result in a waste of power. This is a scenario similar to when high-horsepower engines are installed in small cars or cars of substandard quality, or when low-horsepower engines are installed in large cars. The use of high-power agricultural machinery entails high rates of energy consumption. For small areas of or for scattered farmland, the energy consumption is not proportional to the working area. The unreasonable planning of operations will lead to idleness or wastes of time, both of which will affect operational efficiency.

Wang Pei et al. [4] studied the correlation between the working area for agricultural machinery and the power of agricultural machinery. They calculated the area, time, and efficiency of a subsoiling operation, and established a benefit model for the subsoiling operation based on tractor power through linear regression analysis, proving that there is a clear linear relationship between tractor power and the benefit of a subsoiling operation. However, the quality of an agricultural machinery operation is not only related to the area worked per unit of time, but also to fuel consumption, operation quality, transition time, etc. We should not blindly increase the power of agricultural machinery to improve operational efficiency.

Lavrov A.V. and Shevtsov V.G [5–8] studied the development of the tractor market in Russia from 2008 to 2014 and evaluated tractor levels using an improved method of evaluating local production levels for agricultural mobile energy products, taking the fleet as a whole and evaluating the potential efficiency of the fleet according to extensive statistical data on the dynamic changes in the main crop production indicators.

In 2011, the American Association of Agricultural and Biological Engineers put forward two standards to measure the efficiency of agricultural machinery operations [9], namely, effective field capacity (EFC) and field efficiency (FE). It is simple and effective to use high-power tractors to pull wide implements to improve the operating capacity of agricultural machinery [10]. Due to the high power and fast driving speed of the tractor, the daily working area also increases. Although high-power agricultural machinery allows for a large area to be worked every day, oil consumption increases accordingly, and operational efficiency cannot be measured by operating capacity.

Analysis of the efficiency of agricultural machinery operations focuses on the area of operation per unit of time, while the analysis of operational benefits considers the comprehensive benefits brought about by various factors. The benefit of a given agricultural machinery operation is also affected by the length of the transition time. In order to increase the benefit of an operation, one ought to reduce the transition time and distance covered. As the proportion of operation tracks in a plot increases, the benefit of the agricultural machinery operation will also increase. Researchers focus on agricultural machinery scheduling and multi-machine collaborative operations [11–15] in order to maximize the optimization of agricultural machinery task allocation, bolster mutual cooperation and improve the utilization of agricultural machinery. The task of intelligent agricultural machinery operation path optimization has effectively improved the benefits generated by farmland operations and reduced the costs of agricultural machinery operations, both of which are particularly important in the case of large-scale cooperatives, which have many plots and wide service scopes.

Considering one plot, the effective working area is the working area of the plot, and headland turning, repeated operations, etc., are invalid operations that increase fuel consumption and time costs. There are many factors that affect the quality of an agricultural machine's daily operation, including operational efficiency, fuel consumption, manipulator proficiency, failure rate, operation quality, loss of generated data, etc. Studying the daily operation efficiency of agricultural machinery can provide a basis for agricultural machinery managers to evaluate their satisfaction with daily operations so as to improve operation modes and management levels. This paper screens various operational factors according to the operational processes involving agricultural machinery, and the factors of repetitive operation rate are selected as evaluation indicators. These include operation time, operation area, fuel consumption, operation quality, repeated operation rate factors, proportion of effective working time and so on. A particle swarm optimization algorithm is used to improve the LSSVM training model to estimate working efficiency. To solve the problem of large quantities of data and difficult-to-label samples, a semi-supervised learning method is used to increase the number of training samples and improve the accuracy of the training model.

## 2. Analysis of Agricultural Machinery Operation Data

The operating parameter of one agricultural machine in one day is a piece of data. We classify the operation data of 5000 sets of agricultural machinery according to horsepower and calculate the daily average operation area of tractors with different horsepower, as shown in Figure 1.

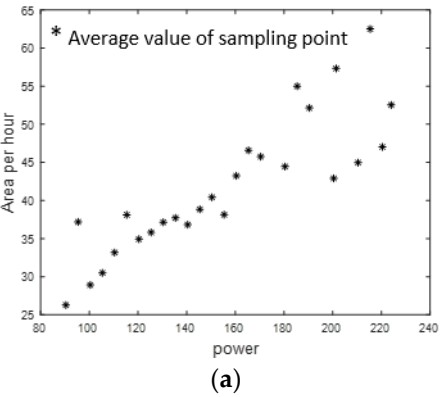
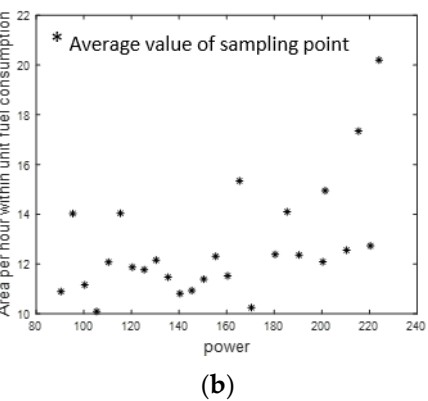

(**a**)  (**b**)

**Figure 1.** Operation effect picture of tractors with different horsepower. (**a**) Relationship between power and working area; (**b**) Relationship between power, fuel consumption and working area.

The abscissa in Figure 1a is the horsepower of the tractor, and the ordinate is the working area per hour, i.e., the working efficiency. It can be seen from the figure that the working efficiency and horsepower are generally proportional and linear. When the fuel consumption factor is increased, as shown in Figure 1b of the figure, the abscissa is the power, and the ordinate is the working area per hour under the unit fuel consumption. There is no obvious linear relationship between the working efficiency and the power of agricultural machinery.

Therefore, when evaluating the one-day operation efficiency of agricultural machinery, the results will be different due to different emphasis and inspection indicators. If only considering the working area in unit time, the greater the horsepower is, the higher the working efficiency and satisfaction will be. We see the results when the factor of working time is increased in Table 1.

**Table 1.** Effect of Different Operation Duration of Agricultural Machinery with the Same Power.

| Power (Horsepower) | Machine Width (mm) | Operation Area (mu) | Duration (min) |
|---|---|---|---|
| 210 | 3900 | 15.16 | 33 |
| 210 | 3900 | 59.48 | 123 |
| 210 | 2400 | 152.79 | 337.2 |

The three agricultural machines have the same power and similar working area per unit time, but the third agricultural machine has a longer operating time and a higher utilization rate of the agricultural machine, so the benefit is higher. Similarly, as shown in Table 2, when pieces of agricultural machinery with different levels power have completed an approximate area of work, high-power tractors will have a larger area of work per unit time and a higher efficiency. However, low-power agricultural machinery can complete the same operation area in the case of taking a lot of time every day, which shows that the utilization rate of agricultural machinery is high, and the operating efficiency is not lower than that of high-power tractors.

In deep sub-soiling operations, the sub-soiling depth exceeds the threshold before reaching the standard. Only when the rate of reaching the standard in the plot reaches more than 95% can it be regarded as qualified. As shown in Table 3, although some agricultural machinery has a large area and high efficiency in deep sub-soiling operations, the operation

quality is poor. This leads to unqualified operation and affects the acceptance, causing more losses to the agricultural machinery owner.

**Table 2.** Effect of completing the same working area with different types of agricultural machinery.

| Power (Horsepower) | Machine Width (mm) | Operation Area (mu) | Duration (min) |
|---|---|---|---|
| 90 | 2600 | 308.75 | 781.2 |
| 130 | 3900 | 324.62 | 286.2 |
| 150 | 3900 | 329.57 | 290.4 |

**Table 3.** Different operation quality of different types of agricultural machinery.

| Power (Horsepower) | Machine Width (mm) | Operation Area (mu) | Pass Rate (%) |
|---|---|---|---|
| 200 | 3600 | 76.09 | 84 |
| 140 | 3900 | 172.73 | 45 |
| 200 | 3600 | 191.65 | 69 |

In addition, due to the driver's technical problems, repeated operations occur during the operation. As shown in Table 4, although the operation area and qualification rate of the three types of agricultural machinery are high, the repeated operation area is also high. The more work is repeated in the same area plot, the more fuel consumption and time cost increase.

**Table 4.** Repeated operation of different types of agricultural machinery.

| Power (Horsepower) | Machine Width (mm) | Operation Area (mu) | Area of Repeated Operation (mu) |
|---|---|---|---|
| 130 | 2600 | 245.5 | 48.46 |
| 200 | 3600 | 155.07 | 64.55 |
| 180 | 3900 | 683.48 | 93.97 |

To sum up, the daily operational benefits does not depend on one or several indicators, and the factors affecting the operation should be selected for comprehensive judgment.

### 3. Indicator Selection

Although indicators, such as failure rate and seedling emergence rate, can also measure the performance and operation of agricultural machinery, they cannot evaluate the operation of a certain day. To judge the daily operation efficiency of an agricultural machine, in addition to the working area per unit time, other factors that have a greater impact and can be obtained every day should also be considered as evaluation indicators. This paper selects daily working area, daily fuel consumption, fuel consumption per unit area, proportion of effective working time, working quality, proportion of effective working area, etc. as evaluation indicators.

Working area (S): As the most important indicator for judging the operating efficiency of agricultural machinery, the daily working area of agricultural machinery is also the standard for measuring the performance of agricultural machinery. The daily operating area is the effective working area, determined by removing overlapping working area, and the operating area is proportional to the power of agricultural machinery, which is the basis for fee collection, subsidy application and wage payment.

Fuel consumption (FC): Agricultural machinery will consume diesel oil in the process of driving on roads and in fields. Agricultural machinery with large power will also consume more fuel per unit time. The lower the fuel consumption as the main cost, the greater the area of agricultural machinery operation, and the higher the efficiency of agricultural machinery operation. With the same working area, the fuel consumption

is the basis for testing the driver's skills, as well as the tractor's performance and operating efficiency.

Work quality (WQ): Deep sub-soiling operation is qualified only when the proportion of the operation area with a depth of 25 cm in each plot exceeds 95%. The quality of deep loosening operation is shown in Figure 2. The red dot in the figure represents the locus point with a depth of more than 25 cm, and the green dot represents the locus point with depth of less than 25 cm. During field operation, the smaller the proportion of green dots is, the higher the qualification rate of deep sub-soiling operation will be.

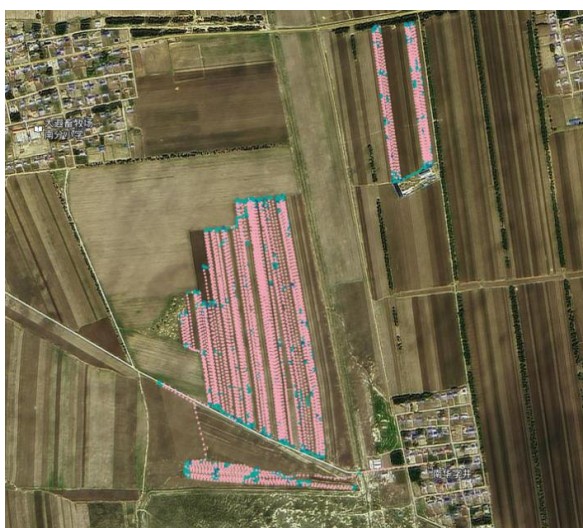

**Figure 2.** Track point diagram of agricultural machinery deep loosening operation.

Repetitive operation rate (RP): Due to unqualified operation, driver negligence and other reasons, repeated operation will occur in the plot. This will not only cause waste of time and fuel consumption, but could also be deemed as malicious compensation fraud if the area of repeated operation is too large in the process of subsidy distribution and evaluation. The repetition rate is shown in Equation (1).

$$\text{RP} = \frac{S_R}{S - S_R} \times 100\% \tag{1}$$

where: $S$ represents the working area; $S_R$ represents the area of repeated operations.

Missing operation rate (MP): Due to terrain, sensor data loss or driver negligence, there will be one or more lines of track vacancy in the operation track. It will not only affect the operation acceptance, but also affect the crop yield in the plot. Missing operation rate is shown in Equation (2).

$$\text{MP} = \frac{S_M}{S - S_R + S_M} \times 100\% \tag{2}$$

where: $S_M$ represents the area of missing work.

The smaller the area of missed work in the plot is, the better the operation effects should be. Because of the gap between crops, the distance threshold should not be less than the ridge spacing when judging the area of missing crops.

Proportion of effective operation time (TP): The ratio of effective operation time is the proportion of field operation time and daily driving hours. When agricultural machinery is operated in contiguous plots, the road driving distance is short, the proportion of operation time in the plots is large, and the utilization rate of agricultural machinery is high. On the contrary, if the two plots are far apart, the road driving distance is long, and the utilization rate of agricultural machinery is low. As much is shown in Figure 3.

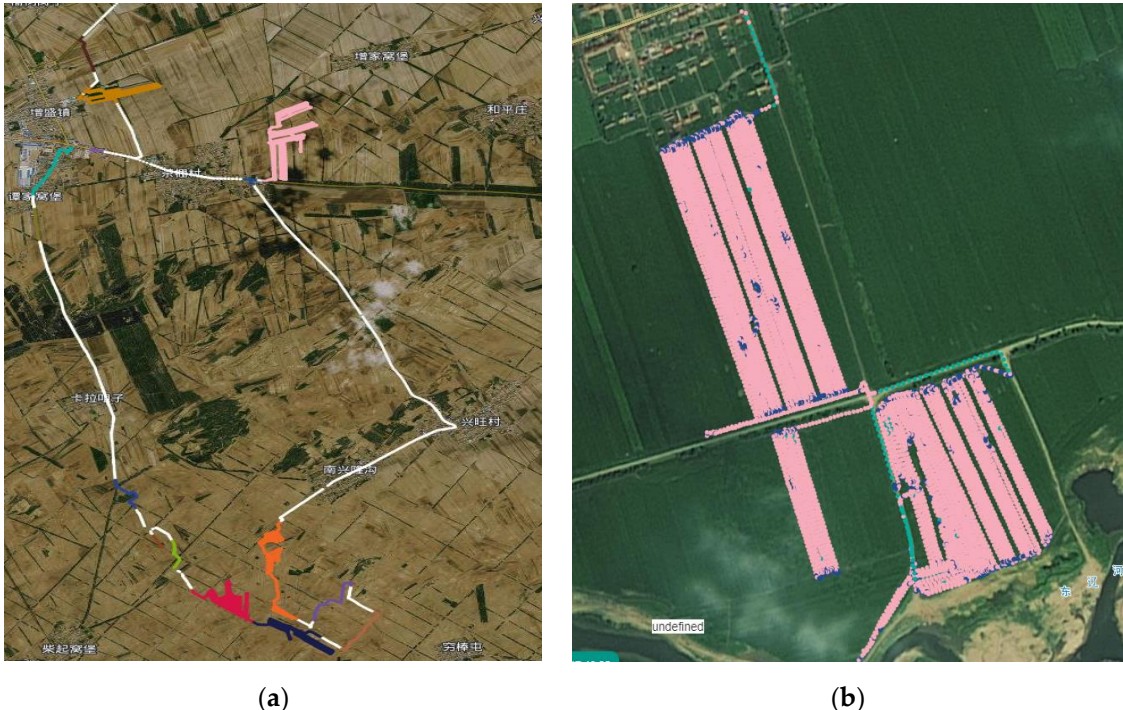

(**a**)　　　　　　　　　　　　　　　　　　　　　　(**b**)

**Figure 3.** Operation track diagram of agricultural machinery (**a**) Long transfer distance and scattered plots; (**b**) Close transfer distance and concentrated plots.

In Figure 3a, the plots are scattered, the area is small, the transfer distance is far, and the proportion of effective operation time is small; The plot area in Figure 3b is large and concentrated, so it does not require too much of a transition time. Therefore, the driving time is mainly used in field operations, and the effective operation time is relatively high. In addition, the turnaround time and parking time are invalid operation times. The proportion of agricultural machinery operation time is shown in Equation (3).

$$\text{TP} = \frac{T_w}{T} \times 100\%　\text{(3)}$$

where: $T_W$ is the effective working time in the field; $T$ represents the total driving time of agricultural machinery.

The greater the proportion of agricultural machinery operation time, the more effective the time of field operation recorder, and the higher the operation efficiency.

The daily data and indicators of each agricultural machinery are counted as the bases for operational benefits evaluation. The daily operation is scored manually, the data is trained, and the operation is estimated with the training model, which is used to guide the agricultural machinery management department in personnel assessment, agricultural machinery evaluation, operation scheduling, etc.

## 4. Operational Benefits Evaluation

According to the data indicators, work efficiency is scored manually. Because the scoring standard is unclear and subjective factors are too large, the accuracy cannot be guaranteed. Compared with scoring, the operation effect is divided into good, medium and poor categories, which are marked manually to reduce the requirements for staff and make it easier to operate. The labelled samples are trained by the classifier. After the training model is established, the test data are classified.

Through regression estimation, we can fit the distribution rule of samples and score the data linearly. This not only reduces the occurrence rate of sample marking errors caused by subjective factors, but also provides more abundant results for reference. In this study, the

LSSVM [16–18] proposed by Suykens et al. starts with the machine learning loss function, uses two norms in the objective function of its optimization problem, and uses equality constraints instead of inequality constraints in the standard algorithm of Support Vector Machines (SVM). The result is that the solution of the optimization problem of the LSSVM method becomes the solution of a set of linear equations. The trajectory point data are predicted by LSSVM regression, and the regression line is obtained as shown in Figure 4.

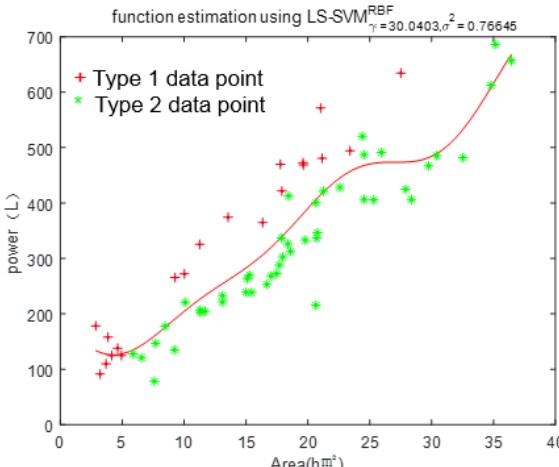

**Figure 4.** Rendering of LSSVM processing operation area and fuel consumption data.

It can be seen from the figure that, although the accuracy of data classification by a regression line is lower than that of SVM classification, the trend of the regression line is more consistent with the data distribution pattern. By modifying the classification standard, the interference caused by initial manual classification is reduced. Therefore, LSSVM is selected to establish a regression model to evaluate the performance of agricultural machinery.

### 4.1. Improved LSSVM by Particle Swarm Optimization

LSSVM is used to solve and realize the decision function. Compared with SVM, LSSVM reduces the difficulty of solving and improves the speed of solving, which is suitable for solving large-scale problems in general applications. However, since the initial parameters are given, generally only the local optimal solution can be obtained, but not the global optimal solution, and so a certain degree of accuracy is sacrificed. As the amount of data increases over time, the time consumption also increases. The calculation time changes with the amount of data, as shown in Figure 5.

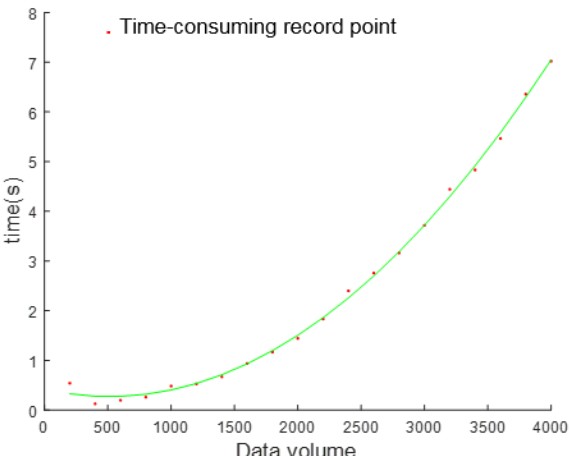

**Figure 5.** Time consumption of LSSVM algorithm changes with data volume.

In the figure, LSSVM is a fixed parameter, i.e., the time consumed for one calculation. When the number of search parameters is increased, the calculation time also increases. To improve the accuracy and stability of LSSVM, it is necessary to determine the value of super parameters in the modeling process. Two key super parameters, γ and σ, are regularization parameters and kernel function parameters. These two parameters have a key impact on the complexity, accuracy and stability of the model, and so it is necessary to optimize them in advance. If the width parameter σ of the kernel function is too large, the support vector will be reduced and the samples will be under-trained, which will simplify the model and reduce its accuracy. If the size is too small, the model will be over-saturated, and the sample will be over-trained. Regularization parameter γ, also known as the penalty coefficient, can minimize the model structure risk by adjusting the ratio between confidence range and empirical risk. When the γ value is small, the empirical risk is relatively weak in the parameter optimization principle based on structural risk minimization. The complexity of the model will be reduced, and the accuracy will be poor, resulting in the model not learning. When γ is large, the structural risk focuses on the empirical risk items, and the model accuracy will be improved, but the model will be complex, and there will be overlearning.

Therefore, how to determine the kernel function parameters and regularization parameters is the key problem in building the LSSVM model. The optimization of σ and γ has a great impact on the complexity, prediction accuracy, stability and generalization ability of LSSVM. For parameter selection of LSSVM, the exhaustive method is adopted. The exhaustion method is algorithmically simple but inefficient, large in computation, and depends on the initial empirical value setting. For parameters γ and σ, the general scope of γ is [0.1300], while the general scope σ is [0.1100], and the calculation is performed in steps of 0.1. To enumerate all parameter possibilities, many calculations are necessary. If the number of samples is large, the time cost of exhaustively finding the optimal parameter is too large. Given a set of empirical values, the calculation speed can be improved by limiting the number of iterations or parameter changes, but the global optimal solution may not be obtained.

In this study, the optimization capability of particle swarm optimization (PSO) is selected to optimize the parameters of LSSVM [19–21] so as to further improve the model performance. The PSO algorithm updates its own speed and position after each iteration according to the optimal solution of the particle itself and the global optimal solution. Each particle cooperates with each other to find the fastest direction and optimal solution. The search speed is fast because only the optimal particles pass information to other particles during the iteration.

The PSO algorithm first determines the position, velocity and direction of particles. After initializing the position and velocity of a group of random particles, all particles are searched in a D-dimensional space.

The algorithm is initialised as a group of random solutions, and the optimal solution is found through iteration. In each iteration, particles update themselves by tracking two "extreme values". The first one is the optimal solution found by the particle itself. This solution is called the individual extreme value, pBest. The other extreme value is the optimal solution found by the whole population. This extreme value is the global extreme value, gBest. To improve the convergence speed, it is also possible to use only the neighbors of some of the best particles instead of the whole population, so that the extremum in all the neighbors is the local extremum.

We randomly generated a D-dimensional search space, with an m particle community. The position of particle μ is expressed as a vector. The particle velocity is v, the individual optimal position is pBest, and the global optimal position is gBest. The PSO algorithm optimizes the parameters to obtain the optimal solution. In the basic PSO algorithm, the iterative formula for updating the speed and position of the individual extremum and the global extremum in the *k*th iteration is:

$$V_{id}{}^{k+1} = \omega V_{id}{}^{k} + c_1 r_1 (P_{id}{}^{k} - X_{id}{}^{k}) + c_2 r_2 (P_{\alpha d}{}^{k} - X_{id}{}^{k}) \tag{4}$$

$$X_{id}^{k+1} = X_{id}^{k} + V_{id}^{k+1}\mathrm{E} \tag{5}$$

where, $\omega$ represents inertia weight; $d$ represents the D dimension of the velocity and position; $V_{id}^{k}$ represents particle velocity; $X_{id}^{k}$ represents the current position; $c_1$ and represent acceleration factors; $r_1$ and $r_2$ represent random numbers between (0, 1); $P_{id}$ represents the historical optimal solution of particle $i$; $P_{\alpha d}$ represents the historical optimal solution of the particle swarm.

The fitness value of the function is accuracy. In each iteration, the local optimal solution and the global optimal solution of each particle are calculated. If the fitness is greater than the last one, the position this time is taken as the optimal solution. The iteration end condition can be set as the number of iterations or the incremental change threshold. At this time, the fitness function value corresponding to the optimal solution of all particles in the population is the largest among the historical values.

First, the optimal position of each particle and the overall optimal position are defined as a matrix; next, the learning factor is initialized to 2, and the hyperparameters $\gamma$ and $\sigma$ are initialized. The parameters are substituted into the LSSVM algorithm to calculate the fitness of each particle. When the particle fitness is greater than the current optimal fitness, the particle fitness and position are updated; when the fitness is greater than the optimal fitness of the population, the optimal fitness and location of the population are updated. When the number of iterations reaches the set value or the fitness does not change, the algorithm ends and the super parameters $\gamma$ and $\sigma$ are obtained.

Taking the daily operation area and fuel consumption as the evaluation characteristics, 400 operation data are selected and marked as two categories, with values 1 and $-1$, 200 of which are used as training sets and 200 as test sets. The prediction results are shown in Figure 6.

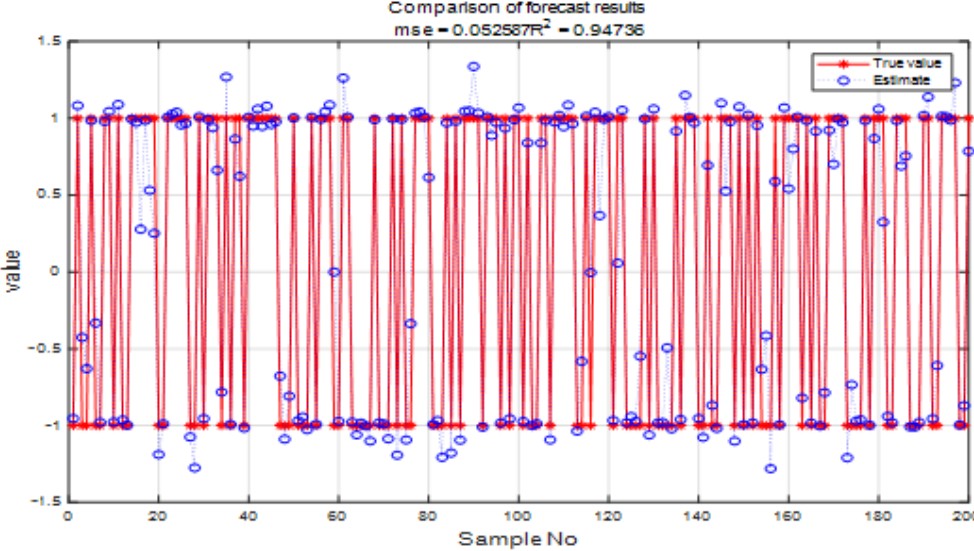

**Figure 6.** Comparison between predicted value and true value of test sample.

As can be seen from the figure, the LSSVM method can effectively distinguish, and the predicted value floats near the real value. It shows that this method is effective for a variety of scoring standards.

*4.2. Semi-Supervised Learning Algorithm Training Model*

There are a large number of agricultural machinery operations in the operation season. Agricultural machinery operation information is automatically collected through airborne equipment. According to the indicators required for operation evaluation, one daily operation of an agricultural machine is one piece of data. To evaluate the efficiency of operations, it is necessary to manually classify some data to generate labels. Due to the

large subjective factors, when the amount of data is small, people can concentrate on strict classification according to the operation. When the amount of data is too large, and not only time-consuming and laborious, manual classification will appear random or wrong due to fatigue. Using a small number of samples to build a model to predict a large number of samples will affect the accuracy of the training model due to insufficient training samples. Although increasing the number of samples can improve the accuracy, it also increases the time cost and difficulty of sample marking. The learning method that combines a small amount of labelled data with a large amount of unlabelled data is called semi-supervised learning [22–24]. This entails using a small number of labelled samples to train the model, continuously selecting data from unlabelled samples to add labelled samples, and increasing the number of labelled samples to train the model, which can not only increase the accuracy of model training, but also reduce the cost of sample labelling.

Supervised learning labels all training samples; this requires a large number of training samples to improve the accuracy and generalization ability of the model, and it is time-consuming and laborious to mark the samples. Unsupervised learning does not need to label samples, but the obtained model is not accurate enough. Semi-supervised learning is between the two, including a small number of labeled samples and unlabeled samples. Increasing the number of labeled samples during training can achieve better results than unsupervised learning. Increasing the number of labelled samples in the training process can achieve better results than unsupervised learning. Many studies have shown that unlabelled data is indeed useful for learning. At the same time, the use of labelled data and a large quantity of easily accessible unlabelled data enriches the training sample information, and the subsequent learning results are better than those of only using a small number of labelled data.

The self-training algorithm [25] (Self Training) is selected in this study, and its working principles are as follows:

1.　Mark a part of the data set, and divide the marked data into training samples and test samples;
2.　The marked training samples are trained to obtain the training model;
3.　The training model is used to predict all unlabelled samples;
4.　Set the top N prediction tags with the highest score and accuracy above a certain threshold as "pseudo tags";
5.　Put the samples with "pseudo tags" into the training sample set, and combine the new training sample set to retrain the model;
6.　Repeat steps 2–5 until the training samples are no longer increased;
7.　Use the generated training model to predict the marked test samples.

After the labelled samples are trained to attain the model, the unlabelled samples are predicted, and the unlabelled samples with correct rates above a threshold are labelled with class tags and put into the training samples. To avoid the degradation of classifier performance, a higher threshold should be set to reduce the generation of false labelled samples. Although the reliability of labelled samples can be improved, mislabelling is inevitable. Therefore, the maximum class spacing method is used to screen this cohort of data after some unlabelled samples are generated as "pseudo labels". The steps are as follows:

1.　Calculate the maximum class spacing Lmax of the original sample set;
2.　Calculate the maximum distance L from a pseudo label sample to the original sample;
3.　If L ≤ Lmax, put the sample into the original sample set; otherwise, put the sample back into the unlabelled sample set;
4.　Update Lmax and repeat steps 2–4 to screen all newly generated samples;
5.　Retrain the model with a new sample set.

## 5. Benefit Evaluation Verification and Method Application

A total of 33,000 agricultural machinery operation data in 2021 are selected for the experiment. The daily operation information of each agricultural machinery is a piece of data, including agricultural machinery model, machine width, daily operation area, oil consumption, operation quality, repeated operation rate, missed operation rate, operation duration, and the proportion of effective working hours.

### 5.1. Data Preprocessing

1.  Screen out data with incomplete information. Due to the problem of data collection and transmission, part of the data is lost, which will result in incomplete evaluation indicators, and these data need to be deleted in advance;
2.  Delete abnormal operation data. The obvious abnormal data of some indicators caused by manual input errors should be deleted, and the daily operation area of some agricultural machinery is too low for reference, so it needs to be deleted as well;
3.  Cluster data to remove noise points and outliers;
4.  Select representative data for manual scoring, in part as training samples and in part as test samples.

### 5.2. Semi-Supervised Training Model

For different types of agricultural machinery, 1000 pieces of data are manually screened and scored, of which 500 pieces are used as training samples and 500 pieces are used as test samples.

1.  Bp neural network is selected to train the training set model, and error samples are manually checked and relabelled;
2.  Conduct training on unclassified samples, and select the top N with the highest score over 98 to enter the training set;
3.  Repeat step 2 until the number of times reached or the number of training sets does not change;
4.  Test the test set.

Use the PSO+LSSVM algorithm to process six-dimensional data with category labels, where the number of iterations and the changes in fitness values are shown in Figure 7.

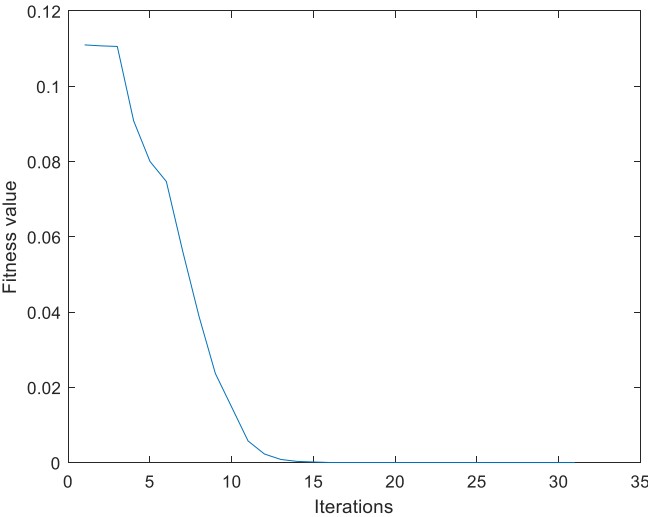

**Figure 7.** Figure of fitness changes with the number of iterations.

The convergence speed is accelerated after 7 iterations, and the model has basically converged when it reaches 25 iterations. The PSO+LSSVM algorithm has faster convergence speed than the LSSVM algorithm alone, and the optimal solution can be found quickly.

In order to verify the effectiveness of the PSO+LSSVM method, this study selects LSSVM, BP neural network and logistic regression methods for comparison. Data manual is divided into three categories. The experimental results are shown in Table 5.

**Table 5.** There are three types of experimental results for the evaluation criteria of operational efficiency.

| Method | Time Consumption (s) | Accuracy (%) |
| --- | --- | --- |
| lssvm | 48 | 92.89 |
| PSO+LSSVM | 26 | 94.43 |
| BP network | 71 | 92.16 |
| Logistic regression | 26 | 91.73 |

It can be seen from the results that compared with LSSVM and neural network algorithms, PSO+LSSVM proposed in this study has faster processing speed and better accuracy than other methods. Due to the great subjective influence in the manual classification process, the use of PSO+LSSVM algorithm to give operation benefits score can help correct the manual evaluation. Choosing the score as the result can provide a better reference basis for the administrator. In addition, the evaluation system designed in this study can select different weights for different indicators according to different personal preferences and set training templates to retrain the model.

The self-learning method is used to train labelled samples, and then the unlabelled samples with high confidence are screened, so as to enrich the training sample set and improve the accuracy of the model. In the self-learning process, to reduce the false labelling of samples, the correct rate threshold is set to 98%. After the unclassified samples are labelled each time, the maximum class spacing method is used to screen out the samples that are "far away" from the samples in the training sample set, and then the qualified samples are added to the training sample set. At this point, the manually labelled dichotomous test is selected. As the number of iterations increases, new unlabelled samples with "pseudo labels" are gradually added to the sample set. The number of new training samples changes with the number of iterations, as shown in Figure 8.

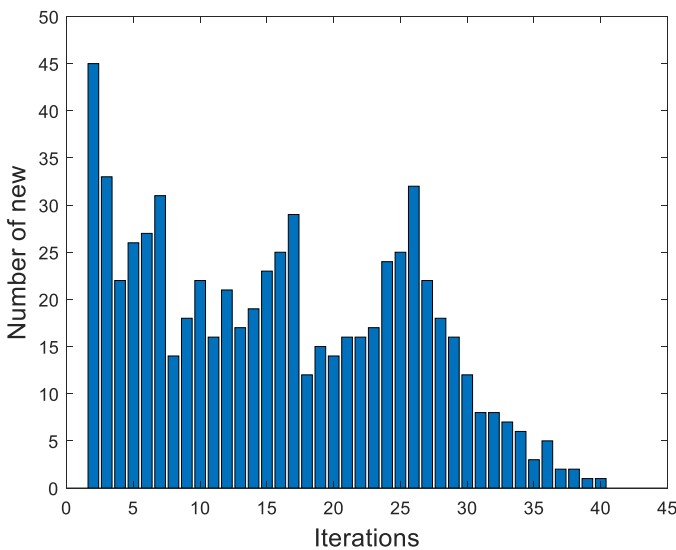

**Figure 8.** Graph of the number of new samples changing with the number of iterations.

With the enrichment of the training sample set, the training model will also change. The model generated by each updated training sample will be used to predict the test sample, and the accuracy change is shown in Figure 9.

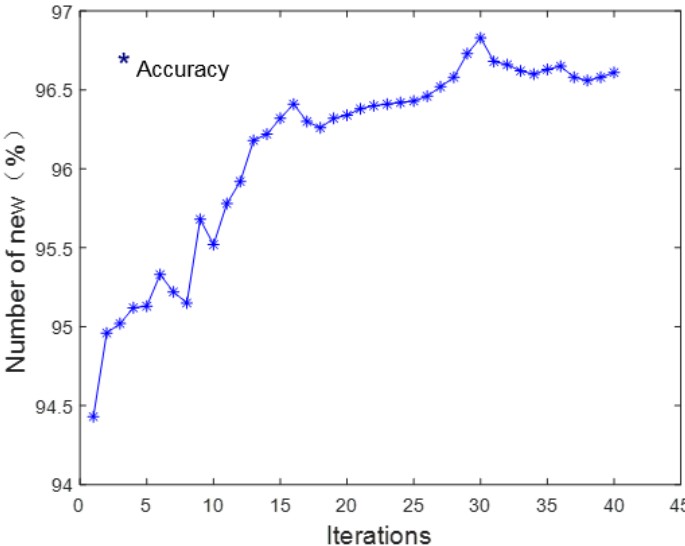

**Figure 9.** Prediction accuracy versus self-learning iterations.

As can be seen from Figures 8 and 9, the number of training samples increases with the number of iterations. At the beginning of the iterations, the number of new samples is large. After 30 iterations, the number of new samples each time becomes small. The prediction accuracy of test samples did not increase with the increase in training samples, but the prediction accuracy increased overall. There are two possible reasons for this result:

1.  The accuracy of the training model does not strictly change with the increase in the number of training samples, but fluctuates;
2.  With the increase in training samples, misclassified samples will inevitably appear, which will affect the accuracy of model training.

In general, the LSSVM regression prediction, improved by the particle swarm optimization algorithm, effectively uses unlabelled samples under the self-learning method, and the generated training model improves the prediction accuracy of test samples.

*5.3. Recommended Combination of Agricultural Machinery and Tools*

In the use of agricultural machinery, many factors will affect the operational efficiency, including unreasonable selection of agricultural machinery, unscientific multi-machine marshalling, poor operation speed control, low time utilization and so on. The pursuit of high-powered agricultural machinery and lack of suitable agricultural machinery will result in a waste of power or in substandard quality. High-power agricultural machinery generates high energy consumption. For small or scattered farmland, the energy consumption is not proportional to the working area. Unreasonable operation planning will lead to idling or waste of time, which will affect operation benefits.

In a deep sub-soiling operation, only when the depth reaches the specified value and the overall qualification rate of the plot reaches the standard can it be regarded as qualified and then receive subsidies; otherwise, the operation is invalid. Therefore, the operation quality or the operation qualification rate is an important indicator that affects the operation benefits of agricultural machinery. We analyze 33,000 pieces of agricultural machinery operation data, grouping them by agricultural machinery model, and checking the operation qualification rate of each type of agricultural machinery with different widths machines. If the qualification rate exceeds 95%, the operation is qualified. Small horsepower tractors cannot be equipped with machines of too large width. When the width of the machine increases, the proportion of qualified operation decreases. Although the qualified operation rate of large horsepower tractors with a small-width machine is relatively large, it does not mean that the smaller width is better, and it will still waste power. Therefore, it is

impossible to judge the optimal combination of power and width of agricultural machinery from the operation qualification rate alone.

Using the above model to evaluate the operational benefits of agricultural machinery, two types of operational benefits are obtained: good and bad. Grouping by agricultural machinery model and viewing the classification distribution of operating benefits of each type of agricultural machinery with different width machines are performed. Figure 10 shows the quantity distribution of good combination quantity and operational benefits with different widths of 1504 pieces of agricultural machinery.

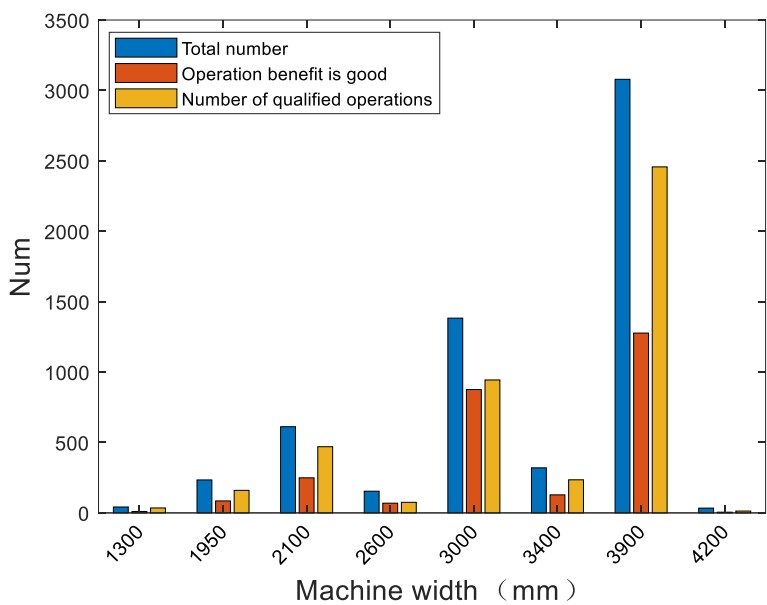

**Figure 10.** Operational benefits histogram of 1504 agricultural machinery with different width machines.

The figure shows that 1504 tractors are equipped with agricultural machinery of different widths for deep sub-soiling operation. The blue color represents the total number of agricultural machinery under this width, the red color represents the number of agricultural machinery with good operation efficiency under this width, and the yellow color represents the number of qualified rate of operation quality above 95%. To increase the area of single line operation, a large quantity of agricultural machinery is equipped with 3900 mm-wide machines, but this leads to a decrease in the qualification rate of the operation. As can be seen from the table, the qualification rate of agricultural equipment with small widths is higher than that of equipment with large widths. When the width is too large, the qualification rate decreases significantly. From the analysis of operational benefits, the proportion of agricultural machinery with 3900 mm-wide agricultural machinery is low. When 3000 mm-wide agricultural machinery is equipped, the proportion operational benefits is highest overall. Therefore, considering all factors comprehensively, it is recommended that 1504 tractor be equipped with 3000 mm-wide machines for combined operation.

Similarly, the optimal combination of agricultural machinery can be recommended for other types of agricultural machinery. The operation effect of some types of agricultural machinery with different widths of machines is shown in Table 6.

The blank space in the table shows the operation data without this combination of agricultural implements. For different types of agricultural machinery, the operational benefits ratio is different when matched with different agricultural implements. The combination with the highest operational benefits ratio can be recommended to management as the best combination.

**Table 6.** Operation effect of different types of agricultural machinery and machines with different widths.

| Width (mm)<br>Power | 1300 | 1950 | 2100 | 2600 | 3000 | 3400 | 3900 | 4200 |
|---|---|---|---|---|---|---|---|---|
| 90 | 0.45 | 0.41 | 0.36 | 0.38 | 0.29 | 0.12 | - | - |
| 110 | 0.32 | 0.38 | 0.49 | 0.33 | 0.42 | 0.31 | - | - |
| 130 | 0.53 | 0.34 | 0.41 | 0.59 | 0.35 | 0.46 | 0.32 | 0.26 |
| 150 | 0.24 | 0.36 | 0.41 | 0.45 | 0.63 | 0.4 | 0.36 | 0.15 |
| 180 | - | 0.38 | 0.46 | 0.38 | 0.66 | 0.51 | 0.35 | 0.43 |
| 200 | - | - | 0.41 | 0.38 | 0.39 | 0.58 | 0.45 | 0.43 |
| 220 | - | - | 0.35 | 0.49 | 0.42 | 0.51 | 0.67 | 0.49 |

*5.4. Analysis of Experimental Results*

The above experiments show that the selected indicators for evaluating the operating efficiency of agricultural machinery are effective. Using the PSO+LSSVM method to build the model can effectively evaluate the agricultural machinery benefits, which is faster than only using the LSSVM method. The semi-supervised method can further improve the accuracy of the model by increasing the number of training samples processed. The experimental results show that, with the increase in the number of training samples, the accuracy has also improved. However, when it reaches a certain level, the accuracy rate does not increase, but decreases. Therefore, setting the sample selection threshold and termination conditions of the semi-supervised method has a great impact on the results.

Using the obtained model to evaluate the efficiency of agricultural machinery operation mainly has the following two advantages. First, it can assess the daily operation effect of agricultural machinery; second, it can also combine the evaluation results to obtain the optimal combination of different types of agricultural machinery through data analysis of different combinations of agricultural implements. This can further guide the matching of agricultural equipment and agricultural machinery operation, thereby improving the efficiency of agricultural machinery operation. According to the experimental results, different types of agricultural machinery may have the same recommended width. As such, the recommended combination of agricultural machinery can only be used as a reference for selection and does not in fact represent the optimal combination.

**6. Conclusions**

(1)  PSO algorithm is used to improve the parameter optimization process of LSSVM. According to its own optimal solution and the global optimal solution, the particle updates its speed and position after each iteration, so as to better find the fastest direction and optimal solution in the iterative process;

(2)  In view of the problems such as large quantities of operation data, time-consuming and laborious sample labelling, and easily made mistakes, a semi-supervised learning method is proposed. A small number of labelled samples are used for training, and a model is generated to predict unlabelled samples. "Pseudo labels" are added to the unlabelled samples whose accuracies are above the threshold. After screening, training samples are added to increase the number of training samples, and to improve the generalization ability and accuracy of the training model;

(3)  Using the LSSVM training model improved by the PSO method, the accuracy of the improved model is increased from 94.43% to 96.83% using the self-learning method, and the optimal combination of agricultural machines is recommended according to the operating efficiency, so as to increase the cooperative efficiency.

(4)  Although the accuracy has improved a little, the model still needs to be optimised. The next step is to give different weights to different indicators, which to increase the scientificity of the model. In the combination recommendation of agricultural machinery, the method used is statistical analysis based on the evaluation results. The next research focus is to find more accurate recommendation methods and obtain more scientific recommendations of agricultural implement combinations.

**Author Contributions:** Conceptualization, Y.L. and B.Z.; methodology, Y.L. and L.W.; writing—original draft, Y.L. and W.Z.; funding acquisition, L.Z.; project administration, B.Z. and L.Z.; formal analysis, Y.L. and W.Z.; investigation, L.W.; software platform, L.W. and L.Z.; data curation and software, visualization, Y.L. and W.Z. All authors have read and agreed to the published version of the manuscript.

**Funding:** The work was sponsored by the National Key R&D Program Project of China (2020YFB1709603).

**Institutional Review Board Statement:** Not applicable.

**Informed Consent Statement:** Not applicable.

**Data Availability Statement:** Not applicable.

**Conflicts of Interest:** The authors declare no conflict of interest.

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
