# Peer review of "Evaluation of Agricultural Machinery Operational Benefits Based on Semi-Supervised Learning"

_agriculture, doi:10.3390/agriculture12122075_

Round 1

Reviewer 1 Report

The issues raised by the authors are relevant and meaningful. The study is innovative and has practical implications for agriculture and could be a starting point for further scientific research. It is hoped that the manuscript will be improved in the following aspects.

1. The method in the manuscript is a combination of semi-supervised methods and other methods, which should be made clear in the summary.

2. The most mentioned in the introduction is operation efficiency, while the manuscript studies operation benefit. Please explain the difference between the two.

3. The introduction should add the significance or objectives of the study.

4. When the abbreviation LSSVM is mentioned for the first time in the manuscript, the full spelling shall be given first, and other similar situations in the manuscript shall be checked.

5. In the manuscript "with 1 and 1200 as the training set and 200 as the test set", what is 1200 here? There is a problem with the original expression, please check and correct it.

6. The conclusions are not very adequate and need to be improved. It is recommended that the conclusion should indicate the shortcomings of the study and the prospects for further investigations.

Author Response

Dear Reviewers,

Thank you for taking time out of your busy schedule to review the manuscript. These comments are all valuable and very helpful for revising and improving our paper, as well as the important guiding significance to our researches. Now we have carefully corrected and replied the manuscript for this revision. The revision instructions are as follows:

 Comment 1. The method in the manuscript is a combination of semi-supervised methods and other methods, which should be made clear in the summary.

 Response 1. Thank you for pointing out this problem. This study uses semi supervised method combined with LSSVM method, which has been modified and explained.

Comment 2. The most mentioned in the introduction is operation efficiency, while the manuscript studies operation benefit. Please explain the difference between the two.

 Response 2. We thank the reviewers for their careful reading of the manuscript and their constructive remarks. In the introduction, the explanation is added: agricultural machinery operation efficiency focuses on the area of operation per unit time, the operation efficiency is considered more comprehensively, and more factors need to be measured

Comment 3. The introduction should add the significance or objectives of the study.

 Response 3. Thank you very much for your suggestions. We have added research significance to the paper, which makes the paper more complete

 Comment 4. When the abbreviation LSSVM is mentioned for the first time in the manuscript, the full spelling shall be given first, and other similar situations in the manuscript shall be checked.

 Response 4:Thank you very much for pointing out this error. When we first mentioned LSSVM in the article, we added Quanpin, checked other similar situations in the full text, and modified the paper writing

 Comment 5. In the manuscript "with 1 and 1200 as the training set and 200 as the test set", what is 1200 here? There is a problem with the original expression, please check and correct it.

 Response 5:Thank you for pointing out this writing error. The correct writing should be “market as two categories, with 1 and - 1, 200 of which are used as training sets and 200 as test sets”. It has been modified in the paper.

 Comment 6. The conclusions are not very adequate and need to be improved. It is recommended that the conclusion should indicate the shortcomings of the study and the prospects for further investigations.

 Response 6. Thank you for your valuable comments. In the conclusions of the paper, the research deficiencies and the next research focus have been added. The focus is to give different weights to different indicators and increase the scientificity of the model..

We sincerely thank you for your warm work and hope that ‎correction ‎will be approved. ‎Thank you again for your comments and suggestions. ‎

Sincerely yours,

Zhao bo

Corresponding author: Zhao bo at Chinese Academy of Agricultural Mechanization Sciences, 100020, Beijng, China, [email protected], phone number: +010-64882659

Reviewer 2 Report

The publication is devoted to the development of a mathematical method for selecting the optimal fleet of vehicles. The application of the proposed Semi Supervised Learning is new and original for this problem. However, there are a few remarks to the presented material.

1. A generalized mathematical model for evaluating the effectiveness of agricultural machinery depending on the specified indicators is not presented: productivity, shift ratio, quality of work performed, etc.

2. In addition to the depth of plowing, other indicators should be added to the indicators of the quality of the work performed: rowing, evenness, etc. It is recommended to consider other technological operations (inter-row cultivation, sowing, etc.) and quality indicators of these operations (seed placement depth, crumbling, etc.).

3. The harmful effect on the soil does not depend on the power of mobile equipment. It is necessary to clarify this statement.

4. There are no references in the work to Russian scientists working on this topic, for example, Lavrov A.V. and Shevtsov V.G.

4. The structure of the presentation of the material differs from the generally accepted for journals.

Author Response

Dear Reviewers,

Thank you for taking time out of your busy schedule to review the manuscript. These comments are all valuable and very helpful for revising and improving our paper, as well as the important guiding significance to our researches. Now we have carefully corrected and replied the manuscript for this revision. The revision instructions are as follows:

 Comment 1. A generalized mathematical model for evaluating the effectiveness of agricultural machinery depending on the specified indicators is not presented: productivity, shift ratio, quality of work performed, etc.

 Response 1. We thank the reviewers for their careful reading of the manuscript and their constructive remarks. The working area and working time are used as parameters to reflect the productivity; The subsoiling operation is selected as the research object, and the best operation quality evaluation is the qualified rate of subsoiling points. The repeated operation rate and missed operation rate can also reflect the operation quality; Because the research is about daily work evaluation, the parameters that do not necessarily occur every day are not studied as influencing factors. Thank you again for your suggestions.

Comment 2. In addition to the depth of plowing, other indicators should be added to the indicators of the quality of the work performed: rowing, evenness, etc. It is recommended to consider other technological operations (inter-row cultivation, sowing, etc.) and quality indicators of these operations (seed placement depth, crumbling, etc.).

 Response 2. Thank you for your professional advice. As the subsoiling operation does not involve planting, seed placement depth, collapsing, etc. were not taken into account. Repetitive operations and missed operations can reflect whether the operations are uniform, whether the number of lines completely covers the land or whether there are omissions.

Comment 3. The harmful effect on the soil does not depend on the power of mobile equipment. It is necessary to clarify this statement.

 Response 3. Thank you for your professional suggestions. This is something we did not consider properly before, and this expression has been deleted in the paper.

Comment 4. There are no references in the work to Russian scientists working on this topic, for example, Lavrov A.V. and Shevtsov V.G.

 Response 4:Thank you for your professional suggestions. Due to the lack of reference scope, we failed to quote the work of the two scientists. We have added two scientists' research as references in the introduction and references.

Comment 5. The structure of the presentation of the material differs from the generally accepted for journals.

 Response 5:Thank you very much for your meticulous and rigorous work attitude. The structure of this paper is as follows: first, analyze the agricultural machinery operation, demonstrate the correctness of the views in this paper, then select appropriate indicators for modeling, then introduce the methods used, and then prove the effectiveness of the methods through experiments, and finally propose the application of the methods in this paper. Although it is different from generally accepted for journals, the structure used in this article is used to introduce the methods in this article clearly. In addition, we listened to experts' suggestions and added the analysis of experimental results to make the results clearer.

We sincerely thank you for your warm work and hope that ‎correction ‎will be approved. ‎Thank you again for your comments and suggestions. ‎

Sincerely yours,

Zhao bo

Corresponding author: Zhao bo at Chinese Academy of Agricultural Mechanization Sciences, 100020, Beijng, China, [email protected], phone number: +010-64882659
